# Combination of Computational Techniques to Obtain High-Quality Gelatin-Base Gels from Chicken Feet

**DOI:** 10.3390/polym13081289

**Published:** 2021-04-15

**Authors:** José C. C. Santana, Poliana F. Almeida, Nykael Costa, Isabella Vasconcelos, Flavio Guerhardt, Dimitria T. Boukouvalas, Wonder A. L. Alves, Pedro C. Mendoza, Felix M. C. Gamarra, Segundo A. V. Llanos, Sidnei A. Araujo, Ada P. B. Quispe, Rosangela M. Vanalle, Fernando T. Berssaneti

**Affiliations:** 1Department of Production Engineering, Polytechnic School, University of São Paulo, Av. Prof. Luciano Gualberto, 1380, Butantã, São Paulo 05508-010, Brazil; 2Department of Management Engineering, Federal University of ABC, University Mall, São Bernardo do Campo 09606-045, Brazil; 3Federal Institute of Mato Grosso, São Vicente Campus, São Vicente da Serra 78106-000, Brazil; poliana.almeida@svc.ifmt.edu.br; 4Industrial Engineering Post Graduation Program, Nine July University, Vergueiro Avenue, 235/249, Liberdade, São Paulo 01504-000, Brazil; nykaelcosta@gmail.com (N.C.); belinnh1@gmail.com (I.V.); flavioguerhardt@gmail.com (F.G.); rvanalle@uni9.pro.br (R.M.V.); 5Informatics and Knowledge Management Post Graduate Program, Nove de Julho University, Vergueiro Avenue, 235/249, Liberdade, São Paulo 01504-000, Brazil; dtbouk@outlook.com (D.T.B.); wonder@uni9.pro.br (W.A.L.A.); saraujo@uni9.pro.br (S.A.A.); 6Facultad de Ingeniería Ambiental y Sanitaria, Universidad Nacional San Luis Gonzaga de Ica, Ciudad Universitaria, Km 305, Ica 11000, Peru; pedro.cordova@unica.edu.pe; 7Energy Engineering, University of Brasilia, FGA-UnB, St. Leste Projeção A—Gama Leste, Brasília 72444-240, Brazil; fcarbajal@unb.br; 8Chemical Engineering Department and CYMAIDS, Universidad Nacional Pedro Ruiz Gallo, Juan XXIII 391 Street, Lambayeque 14000, Peru; svasquezll@unprg.edu.pe (S.A.V.L.); abarturen@unprg.edu.pe (A.P.B.Q.)

**Keywords:** neural network, chicken feet, sensorial quality, food quality, gelatine

## Abstract

With the increasing global population, it has become necessary to explore new alternative food sources to meet the increasing demand. However, these alternatives sources should not only be nutritive and suitable for large scale production at low cost, but also present good sensory characteristics. Therefore, this situation has influenced some industries to develop new food sources with competitive advantages, which require continuous innovation by generating and utilising new technologies and tools to create opportunities for new products, services, and industrial processes. Thus, this study aimed to optimise the production of gelatin-base gels from chicken feet by response surface methodology (RSM) and facilitate its sensorial classification by Kohonen’s self-organising maps (SOM). Herein, a 2^2^ experimental design was developed by varying sugar and powdered collagen contents to obtain grape flavoured gelatin from chicken feet. The colour, flavour, aroma, and texture attributes of gelatines were evaluated by consumers according to a hedonic scale of 1–9 points. Least squares method was used to develop models relating the gelatin attributes with the sugar content and collagen mass, and their sensorial qualities were analysed and classified using the SOM algorithm. Results showed that all gelatin samples had an average above six hedonic points, implying that they had good consumer acceptance and can be marketed. Furthermore, gelatin D, with 3.65–3.80% (*w/w*) powdered collagen and 26.5–28.6% (*w/w*) sugar, was determined as the best. Thus, the SOM algorithm proved to be a useful computational tool for comparing sensory samples and identifying the best gelatin product.

## 1. Introduction

### 1.1. Poultry Production in Brazil

Many countries have been economically influenced by the trade of chicken and despite the international crisis, the production of chicken increased in 2019 with a global poultry production of approximately 98.5 million tons. The three major world producers are USA, China, and Brazil, with 20.6%, 15.1% and 14.2% production, respectively [1]. Brazil was the second largest producer of chicken in 2015, at 13.5 million tons, but in 2019 it was the third largest producer, at 13.3 million tons and it was estimated to reach 14 million tons in 2020. Moreover, Brazil is also the largest global exporter of chicken meat with 4.2 million tons, with an increase of 2% per year. The people of Brazil have been estimated to consume approximately 97.6 kg per capita of meat, comprising 47.3 kg chicken, 36.3 kg beef, and 14.1 kg pork (accounting for 48.5%, 37.2%, and 14.4% of the total, respectively) [1]. However, the Brazilians usually do not consume the chicken by-products (carcasses, chicken feet, viscera, etc.); therefore, these are commonly discarded as industrial waste [2,3].

Hence, with the increased meat production, a significant amount of organic waste is generated in the different stages of the production chain of chicken and beef, in particular, and the residue leads to economic and environmental problems. It is a serious concern as the high organic matter content in these wastes can serve as sources for proliferation of microorganisms [2].

Despite the commercial significance of the waste generated during the slaughter of chickens, previous studies have been limited in identifying other treatment and disposal technologies, which are of concern to the poultry industry in Brazil. Some companies have gradually developed technologies to solve these bottlenecks considering the environmental effects of the production process [4].

### 1.2. Gelatin as a Solution for the Poultry Industry Waste

Gelatin is a denatured fibrous protein derived from collagen by partial thermal hydrolysis. It is an important functional biopolymer that has wide applications in food, material, pharmaceutical, and photography industries [5,6].

The global market for gelatin is estimated at 300 thousand tons per year accounting for more than $2 billion per year of the economy worldwide. Brazil exports approximately 25 tons per year which accounts for 80% of the total production, considering growth in the gelatin market, because of its notable competitive advantages in domestic animal production [7]. Thus, food and pharmaceutical industries worldwide have faced an increasing demand for collagen and gelatin. The most popular and commonly used is the gelatin of mammals (pigs and cattle), which are subjected to greater restrictions and scepticism among consumers due to socio-cultural and health concerns [4,6,8]. In this context, Prosekov and Voroshilin [9] stated that the optimisation of gelatin production technologies and investigation of new raw material sources for its production will contribute to solving the problem of substituting the import of this product and stimulating its export.

The increasing demand for new gelling agents to replace the gelatin of mammals has guided several studies on different raw materials, such as the gelatin of marine origin (fish skin, bone, and fins) [5]. Moreover, other research focussing on the extraction and classification of gelatin from fish have been conducted by [5,10,11,12]. However, it is an underused source, and with the increasing world population, it is necessary to explore new suitable alternatives to meet the increasing demand. Furthermore, these alternative sources should not only have nutritive value and ability to be produced at a large scale with low cost, but also present good sensory characteristics [4,13].

The materials considered as waste in some regions may be the raw materials for traditional products with high benefits in other regions. For example, in some Asian countries, chicken feet are a delicacy, but in Brazil, these are not generally consumed. The cost of chicken feet is below 0.5 US$/ton [14]. These characteristics of national market are crucial in defining its low sale price. Thus, the development of a gelatin with good sensorial qualities from this raw material will add value to the production chain of the meat industry [2,15], for example, a Nile tilapia hamburger developed by incorporating whey, collagen from chicken feet, and taro flour [7]; or gelatines, jellies, and biofilms obtained from the collagen of chicken feet [2,4,15,16]. Chicken feet gelatin has also been used to improve the texture of chocolate spread [16].

### 1.3. Application of Algorithms in Sampling

To ensure that the evaluation of samples by the consumers have significance, a large amount of data is required; however, a part of these data is not required for knowledge extraction. Moreover, data are pre-processed to reduce the amount of information and select more relevant attributes. Therefore, various techniques have been developed; for example, Pereira and Sassi [17] used a Hough sets data clustering technique for the data provided by an insurance company, Chen et al. [18] used a fuzzy neural network on a consumer advertising data set, and Pourahmad et al. [19] used a hybrid fuzzy neural network to assess the service quality of an academic library.

De Pelsmaeker et al. [20] used an algorithm based on the fuzzy set theory using the house of quality (HOQ) in a quality function deployment (QFD) study applied to the food industry and evaluated its possibilities and limitations for improving food products. A case study was conducted on filled chocolates, where consumer preferences, processing parameters, and sensory attributes were evaluated. The results revealed limitations in the application and interpretation of HOQ within the food industry and that by using the fuzzy set theory, chain information was incorporated in the HOQ, which established good communication between departments. Thus, the company was able to produce products with high consumer preference and had a high success rate.

Almeida et al. [21] standardised the preparation of Barbados cherry wines by simulated annealing technique. It is a probabilistic search technique that simulates the annealing process of metals, wherein the metal is heated to high temperatures and then systematically cooled in the same order to achieve an equilibrium, characterised by a uniform and stable microstructure. The optimisation results showed that the best conditions were observed at a mass ratio of 1/7.5–1/6 and 28.6–29.0° Brix total soluble solids, from which a sensory acceptance between 7–9 hedonic points were obtained for colour, aroma, and flavour.

According to Santana et al. [22], the statistical techniques of F-test and t-Student test based on six Sigma, data envelopment analysis (DEA), and Taguchi procedures are commonly used in food technology [21,22,23,24]. However, in the last decade artificial neural networks (ANN) have been increasingly applied in the sensorial quality evaluation of food. Among the ANN models, the Multilayer Perceptron (MLP) and Kohonen’s self-organising maps (SOM) are noteworthy. The MLP is a supervised learning model, commonly used for solving nonlinear problems, that learns a specific function by training on a dataset [25]. In addition to the input and output layers, it has one or more hidden layers that allow the network to map input patterns with similar structures for different outputs. The SOM is an unsupervised learning model to produce a low-dimensional (typically two-dimensional) discrete representation of the input space of the training samples, in the form of a topological map [26]. This model comprises of two layers (input and output), and due to its characteristics, SOM can be used as a dimensionality reduction method.

Santana et al. [22] have applied SOM for clustering sensorial data from Barbados cherry wines, demonstrating that this algorithm is a promising assessment technique for the sensorial quality of foods. Alves et al. [27] employed the SOM to classify wine samples by their sensorial attributes. They demonstrated the potential of this ANN type in the sensory classification of food and to replace known statistical techniques.

Liu et al. [28] compared the Arrhenius model and MLP for the quality prediction of rainbow trout (*Oncorhynchus mykiss*) fillets during storage at different temperatures. Based on the obtained results, they concluded that MLP was more effective, and was a potential tool for modelling quality changes of rainbow trout fillets within the temperature range of 270–282 K.

Ouyang et al. [29] proposed a method that combines MLP and adaptive boosting (AdaBoost) algorithm for estimating Chinese rice wine quality. They observed that the proposed method showed superior results when compared with other algorithms, and therefore, appropriate for predicting the sensory quality in Chinese rice wine.

Yu et al. [30] developed a hybrid model combining linear partial least squares (PLS) regression and MLP for prediction of consumer approval scores for ready-to-drink green tea beverages, which obtained a coefficient of determination (R^2^) of 0.875 and a root-mean-square error (RMSE) of 2.41%.

Lu et al. [31] proposed an approach that combines the principal component analysis (PCA) technique with MLP for modelling the effect of vibration on the quality of stirred yogurt during transportation and determined the optimal transportation distances.

Sarkar et al. [32] compared the classification efficiency of SOM, PCA, and hierarchical cluster analysis (HCA) in an analysis of the nutritional value, X-ray diffraction properties, texture, and sensory characteristics for normal ”rasgulla” samples and pineapple with different drying process. The authors observed approximately similar trends in the results obtained by the three compared algorithms.

Notably, there are several applications of computational techniques in the optimisation of production processes and in analysing the quality of food products. However, after an extensive review of literature from international scientific document bases, no reports were found on studies that applied software to assess the sensory quality of gelatines.

This study aimed to optimise the production process of gelatin-base gels from chicken feet by response surface methodology (RSM) and facilitate the assessment of their sensory quality using Kohonen’s SOM. Factors such as collagen and sugar content were varied in a 2^2^ experimental design (DOE) to assess their effects on the sensorial attributes of colour, flavour, aroma, and texture of these gelatines. A Hough assessment of the sensory characteristics was conducted by applying the ANN of SOM on a set with 2000 data.

## 2. Material and Methods

### 2.1. Preparation of Gelatines

The physicochemical analysis of collagen and gelatin processing was developed by a Federal Institute of Mato Grosso, Brazil and chicken feet were acquired from a partner food company. Chicken feet collagen powder was prepared according to the method described by Santana et al. (2020) [4]. For the production of gelatines, the Good Manufacturing Practices (GMP) were adhered to during the entire process until the storage of the final product, to ensure safety and to avoid contamination of the end product. Eight gelatin samples were used in this study, seven freshly prepared from chicken feet and a commercial gelatin (produced from ox hide) to act as a reference to compare the experimental results. The chicken feet were obtained from markets in the city of São Paulo, Brazil. For preparation, the chicken feet were washed, the nails were removed, and washed again with cold water to remove any residue of dirt, cleaned with chlorine (2 ppm active Cl_2_) water, and then cut and stored in a refrigerator. Then, 200 g of chicken feet was placed in a thermal bath of 4% acetic acid solution at 60 °C for 4 h, to extract the collagen [2,4,5,11,14].

### 2.2. Experimental Design

A 2^2^ experimental design was adopted wherein the collagen powder and sugar contents were used as factors that highly influenced the sensorial responses of consumer to each attribute (colour, aroma, flavour, texture, and general aspect), as showed in Table 1 [22,33,34,35,36]. The gelatines were prepared to a volume of 700 mL, in grape flavour. Commercial gelatin-base gel was prepared by the procedure provided by its manufacturer, Dr. Oetker^®^ (São Paulo, Brazil), and was referred to as Gelatin H [2,4].

After conducting the tests, a model was obtained by the least squares method and its fitting was verified using the analysis of variance (ANOVA). Models were tested for the following attributes as responses: colour, flavour, aroma, and texture with sugar content and collagen mass as factors. Optimisation was performed using the RSM in Software Statistica 10.0 for Windows^®^, São Paulo, Brazil, based on the concept proposed by [22,34,35,36,37]. The extracted material was transferred to Petri dishes and placed in a vacuum oven at 55 °C for 12 h. The dried material was then ground to obtain a powder and characterised according to the standards provided by the Association of Official Analytical Chemists (AOAC) [38].

### 2.3. Sensory Analyses

The acceptability of the gelatin samples was analysed by using sensory affective tests and compared with the sensorial qualities of commercial gelatin. Fifty (20 mL each) gelatin samples were served to the 50 consumers in codified plastic cups covered with a thin layer of plastic film, using a monadic presentation and rated using a hedonic nine-point scale. The consumers also registered their purchasing intentions for each sample on the same score sheet, using a nine-point scale. Sensorial characteristics such as flavour, colour, and aroma of gelatines were evaluated. The quantitative experimental research was conducted wherein the standard sample was used for sensorial analysis and random sampling was applied for each of the above attributes using a hedonic scale (1–9 points). This methodology is described in [15,21,24,30,39].

The numerical values in the hedonic scale represented the sensorial responses of the consumer as: 1—disliked very extremely, 2—disliked extremely, 3—disliked regularly, 4—disliked, 5—no perceived difference, 6—liked slightly, 7—liked regularly, 8—liked extremely, and 9—liked very extremely [15,21,24,30]. Furthermore, based on the frequency of responses, the sensorial data were compared by their *t*-values [24,30,39].

### 2.4. Determination of Chemical Properties

#### 2.4.1. Percentage Composition

All measurements were conducted according to the standard methods of the AOAC [15,23,34,38] and results were expressed as the percent weight loss during the drying process. The moisture content of gelatine and chicken feet were analysed by drying at 105 °C for 8 h. The protein and lipid contents in gelatine and chicken feet were determined by the Kjeldahl method and the Soxhlet method, respectively. The total ash content of previously dried samples was evaluated by calcining at 500–600 °C for 4 h. The average of three batches under optimal extraction conditions was considered. To measure the energy contained in the gelatines, it was assumed that one gram of protein and one gram of lipid had 4 kcal of energy each and one gram of carbohydrate had 9 kcal of energy. A texture analyser TA-XT2, Stable Micro System (Surrey, UK) was used for determination of gel strength at 25 °C and 6.67% gelatin content (*w/w*) [4,19].

#### 2.4.2. Spectroscopic Analyses

The elements Mn, Fe, Al, K, Ca, Ti, Mg, Na, Cr, V, Ni, Zn, Pb, Li, Cu, La, Ce, Th, U, Sr, and Be were determined in collagen ash samples by inductively coupled mass spectroscopy (ICP-MS) using the method by [4,11,12,40,41]. Gelatin content was measured by FTIR spectroscopy using a Nicolet iS5 FTIR spectrometer equipped with an ATR/iD3 with an argon horizontal cell (Thermo Fisher Scientific^®^, Waltham, MA, USA) at 16 °C. The spectra were analysed in the range of 400–4000 cm^−1^ and the automatic signals were obtained in 32 scans at a resolution of 4 cm^−1^ against a background spectrum recorded from a clean empty cell at 16 °C [4,11,12,40,41]. The average of 32 batch samples under optimal extraction conditions was considered as the result.

### 2.5. Gelatin-Base Gel Classification by Self-Organisation Maps

An initial prototype was implemented using the ANN technology and the Visual Basic ^®^ software from Microsoft Co., for evaluating the sensorial qualities of seven chicken feet gelatin-base gel samples. The similar samples were classified into groups by SOM algorithm. The sensorial values attributed by consumers to each gelatin-base gel sample were catalogued in a spreadsheet. Then, the SOM algorithm was applied for clustering these sensorial values, creating a map where each region represented a group [22]. 

The working of SOM algorithm can be summarised as follows: when a pattern (a set of sensorial values in this study) is presented to the input layer, a neuron of the output layer (topological map) is selected to represent this pattern through a competitive process. During the training phase, the network increases the similarity of chosen neuron in the output layer and their neighbours to the pattern presented in the input layer. Thus, a topological map is developed wherein the neurons that are topologically close respond similarly to input patterns with similar characteristics.

Sensorial data set have been represented by the expression: $\zeta =\{(c_{i},p_{i})\in Z^{4}\times \{A,B,\ldots ,H\}:\;i=1,2,\ldots ,50\}$ so that $X_{i}=(c_{i},p_{i})\in \zeta$ to combine a sequence of sensorial evaluations, $c_{i}$ (for aroma, flavour, appearance, and texture), from the product, $p_{i}$ by consumer, i. Their similarities are presented in sensorial ζ in a Kohonen network. Furthermore, the neural network has the following configuration: a 1 × 18 enter vector, 5 × 5 topological structure, 500 iterations, and 0.0005 learning rate [22,27].

## 3. Results and Discussion

### 3.1. Gelatin Composition

Figure 1 presents photographs of the gelatin-base gel samples obtained from each of the formulations as shown in Table 1. There are negligible visual differences between the samples with respect to colour quality. There is a slight visual difference of gelatin F compared to the other samples. However, as this is one sample among 50, therefore, this analysis does not reveal anything definite about the sample quality.

Table 2 shows the analysis results of gelatin from chicken feet. The lipid content was found to be less than 7%, indicating a low-fat content (15% of total daily consumption). Protein content in chicken feet was higher than 78%, and 90% of it is gelatin (70.90% of total composition), revealing that this product was very rich in gelatin. Chicken feet gelatin has almost two and half times more gelatin content than cowhide (~35%) [4]. Approximately 99% of the mineral composition comprised the elements Na, K, Ca, Mg, P, and S, which are of great importance for human health. The gel strength of chicken feet gelatin powder was 295 bloom, which was five times higher than that of cow leather gelatin. Additionally, the amount of calories from this gelatin was 376 kcal, which corresponds to 18.8% of the daily consumption requirement (2000 kcal/day). All quality attributes of collagen from chicken feet were superior to those presented by cow leather gelatin (commercial), as already reported in [2,4,15]. This can be further corroborated with [7,9,12,42], which stated that gelatin should be available to trade from other alternative sources, as pork and beef products are not consumed by everyone, such as Islam and Jewish. The low molecular weight peptides formed during prolonged extraction processes possibly form covalent cross-links during the freeze-drying process [12,42,43,44]. This affected the gelatin content from commercial gelatin and reduced one of the primary qualities expected in gelatin. The process used in this study did not have the same effect on chicken gelatin.

### 3.2. Classification of Sensorial Attributes

Table 3 presents the averages of 50 tests obtained after compiling all sensory data derived from responses provided by the consumers. It was seen that the formulation of the gelatin “A”, with lower concentrations of gelatin and sugar obtained the lowest values, and the mean was less than four points on the hedonic scale (disliked slightly). The formulation with higher concentrations of factors, that is gelatin “D”, had the highest average for their sensory qualities, reaching values close to six points on the hedonic scale (liked slightly).

Samples of gelatin-base gel formulation “G” had sensory values represented by over five points on the hedonic scale following those of Gelatin “D”. However, as it is one of the central point triplicates (Gelatines E, F, and G) and samples E and F did not exhibit similar performance, gelatin G cannot be indicated as one of the best. Furthermore, on adding the 150 analyses of the three gelatin samples, the average would be lower than the five points hedonic scale. Nevertheless, the *t*-test statistical evaluation showed no significant differences among all the gelatin formulations. Table 3 also shows the statistical analysis results. Notably, the lack of significant differences among gelatin samples according to the statistical analyses may demonstrate limitations of the statistical technique. This was because the samples had a high variation in their sensory results, including several outliers that were not eliminated from the original data sets of the samples.

Figure 2, Figure 3, Figure 4 and Figure 5 show the clustering results of the sensorial data from Table 3. They show that the samples were perfectly classified into distinct groups by the hedonic values. It may be noted that the groups obtained by SOM algorithm were different from those of the statistical method. This demonstrated the efficiency of SOM algorithm on the sample clustering, as observed in the significant differences among means of samples presented in this table. Figure 2 presents the results obtained for clustering of aroma. After comparison, sensorial values for some gelatin samples were grouped based on their similarity by the SOM neural network and their average values. Three groups were obtained, with commercial gelatin (gelatin H) in the first group with an average of eight hedonic points; gelatines C, D, E, F, and G in the second group with approximately six hedonic points; and gelatines A and B in the third group with a low average of approximately two hedonic points.

Figure 3 shows the results obtained for clustering of the sensorial quality, flavour. Three groups are also shown wherein commercial gelatin (gelatin H) was in the first group with an average more than eight hedonic points; gelatines D, E, F, and G were in the second group with approximately six hedonic points and gelatines A, B, and C were clustered in the third group with approximately two hedonic points.

Figure 4 shows three groups in which gelatines have been classified according to their attribute, appearance. Here, the commercial gelatin (gelatin H) and gelatin D were clustered in the first group with an average higher than eight hedonic points; gelatines C, E, F, and G comprised the second group with approximately six hedonic points; and gelatines A and B were clustered to the third group with an average of approximately two hedonic points.

For the attribute texture, as shown in Figure 5, the gelatin D was grouped with the commercial gelatin H and their average is higher than eight hedonic points; gelatines B, C, E, F and G comprised the second group with approximately six hedonic points; and gelatin A was in a third group with a low average of approximately two hedonic points.

### 3.3. Optimisation of the Gelatin Production

After regression by the least squares method, the models presented by Equations (1)–(4) were obtained. The values of the correlations were approximately close to unity, and though these are not the best as indicated by previous literature, this does not disprove the validity of the models. As noted, all the models showed a hyperbolic dependence of the factors with the responses. Similar models were obtained by Almeida et al. [15,16], when evaluating the dependence of appearance, flavour, and aroma on sugar concentration as factor, and mass of acerola pulp for the wines of the same fruit. The models in their study also did not have a good fit, but their correlation values were approximately close to unity as those obtained in this study.

Equation (1) shows that the influence of concentrations of the parameters (factors) are similar to the responses obtained for the sensory attribute, colour, as it was solely associated with the addition of the dye; all the gelatin samples had similar concentrations of dyes. As shown by Equations (2) and (3), the influence of sugar concentration was more significant than that of the collagen concentration on the sensory attributes of flavour and aroma. Almeida et al. [15,16] showed that sugar has a major influence in the responses for the aroma. Equation (4) shows that there was a higher influence of the concentration of gelatin than that of sugar in the sensory evaluation of the texture of gelatin, which is due to the fact that the higher amount of gelatin (collagen) provided a texture to the more rigid gels, leading to good consumer acceptance. These behaviours have also been observed by previous studies [5,6,11,12].
(1)$$Color=4.86+0.60\cdot x_{1}+0.48\cdot x_{2}+0.18\cdot x_{1}.x_{2}\;(R=0.8139)$$
(2)$$Flavor=4.11+0.39\cdot x_{1}+0.92\cdot x_{2}+0.18\cdot x_{1}.x_{2}\;(R=0.8827)$$
(3)$$Aroma=4.12+0.42\cdot x_{1}+1.16\cdot x_{2}+0.37\cdot x_{1}.x_{2}\;(R=0.9036)$$
(4)$$Texture=4.91+1.67\cdot x_{1}+0.55\cdot x_{2}+0.11\cdot x_{1}.x_{2}\;(R=0.9249)\;$$

Figure 6a–d show the response surfaces generated after optimising the production process of gelatine obtained from chicken feet, under the conditions considered in this study. An overall analysis showed that increasing the amount of sugar and collagen leads to a better product acceptance for all sensory attributes.

Based on the gelatin that presented the best set of sensory qualities, obtained by the SOM algorithm combined with the result obtained by RSM optimisation, a range of sugar and collagen contents can be used to obtain high quality gelatines, for example Gelatin D. Thus, the optimisation indicated that the qualities of gelatin will also be maintained if the gelatin is produced with levels between 0.8 and 1 for both the factors, implying the production of gelatines with sugar content ranging 26.5–28.5 g/100 g and collagen content ranging 3.62–3.80 g/100 g [23,33,35].

From the results obtained in this study the following advantages may be inferred for the gelatines produced from chicken feet over other gelatines: (1) it uses a very low-cost raw material; chicken feet are available for sale at approximately 1.0 R$/ton [14]; (2) it shows good nutritional and sensorial qualities; (3) the product is acceptable for consumption by everyone, irrespective of their socio-cultural beliefs [4,5,11].

Gelatin obtained from a tailing reduces their disposal in landfills, thereby decreasing their environmental impacts, such as the release of wastewaters and soil contamination. This further reduces the release of foul odours and prevents disease caused by pests generated due to its addition to landfills. Additionally, animal skin is very poorly regarded by consumers, which favours the shift from gelatines obtained from these raw materials to that obtained from chicken feet. The adoption of this technology by the meat industry will result in an increase in their profits, making it economically advantageous for industries [44,45,46,47]. 

According to Maceta, P.R.M., Berssaneti [48] and Cardoso et al. [49], considering the increased demand for natural foods and the variety of food choices, producers aim to differentiate their products to gain consumers. Therefore, producers may invest in new products, such as gelatine or collagen from chicken feet, which is considered a nutritive food, leading to increased consumption of the healthy food products, and increasing the quality of life of consumers. Further studies may be conducted to show that the investment in the sensory analysis software based on SOM neural network will result in high economic returns, because it results in innovative products with unique competitive advantages over other products in the market.

## 4. Conclusions

The results showed that SOM is a suitable computational tool to study sensory characteristics of the samples and identify the best gelatin addition to prepare consumer acceptable gelatin-based gels. The developed models show that colour, flavour, aroma, and texture vary hyperbolically with the sugar and collagen contents. Gelatin samples D, E, F, and G had an average of approximately six hedonic points implying that they had good consumer acceptance and were marketable. Gelatin D, with 3.65–3.80% (*w/w*) powdered collagen and 26.5–28.6% (*w/w*) sugar, was found to be the best. In addition, this sample also showed the best sensorial qualities, varying between six and eight hedonic points. The results obtained may have significant contribution to the poultry production chain, because this study explored the utilisation of by-products. The findings of this research may be expanded to the supply chain to particularly include feet gelatin produced from the meat industry, so as to utilise the entire product life cycle. Furthermore, the use of chicken feet to produce gelatin may contribute to sustainability by minimising the waste from the poultry industry and providing an effective alternative through the development of a new product which also adds value to the supply chain of the meat industry.

## Figures and Tables

**Figure 1 polymers-13-01289-f001:**
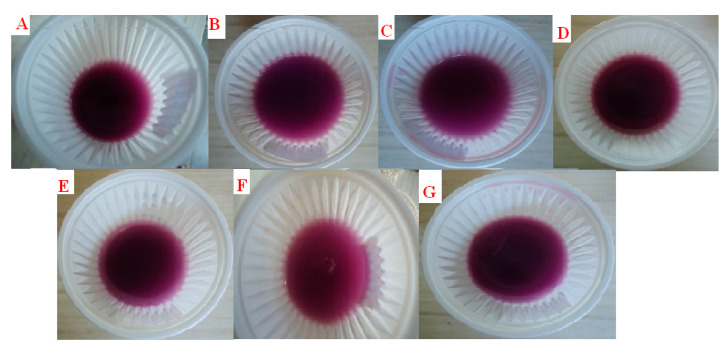
Photographs of the different formulations of gelatin samples produced from chicken feet. (**A**–**G**) gelatin-base gels obtained of according to formulation from experimental design.

**Figure 2 polymers-13-01289-f002:**
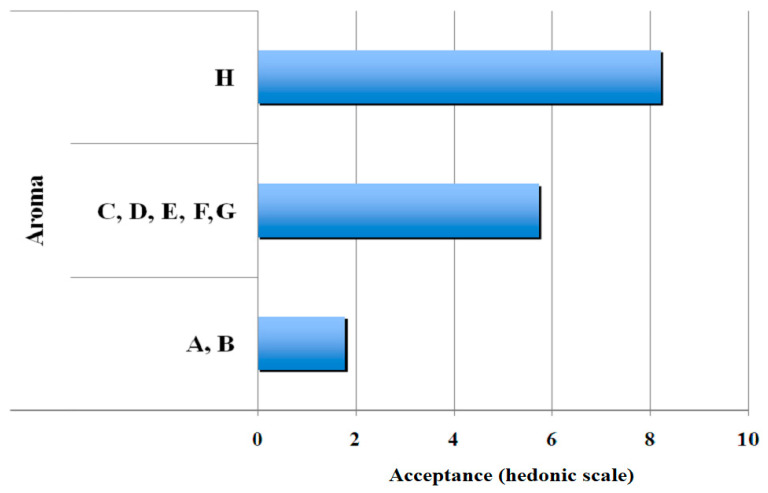
Results of clustering of gelatin-base gel samples for the attribute, aroma.

**Figure 3 polymers-13-01289-f003:**
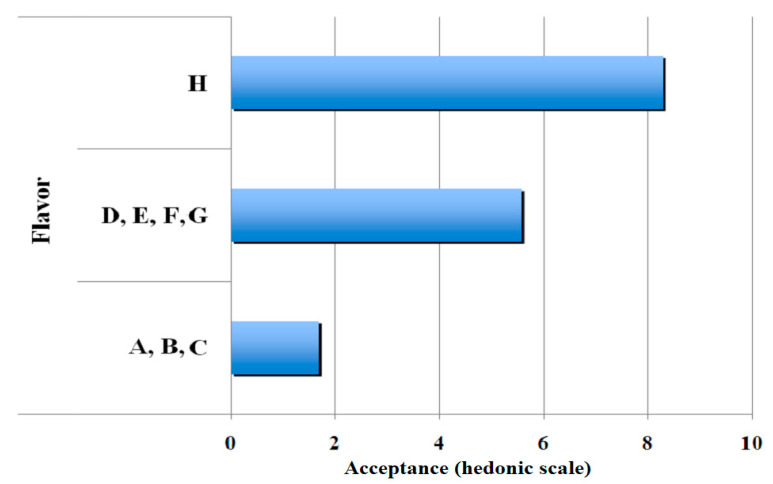
Results of clustering of gelatin-base gel samples for the attribute, flavour.

**Figure 4 polymers-13-01289-f004:**
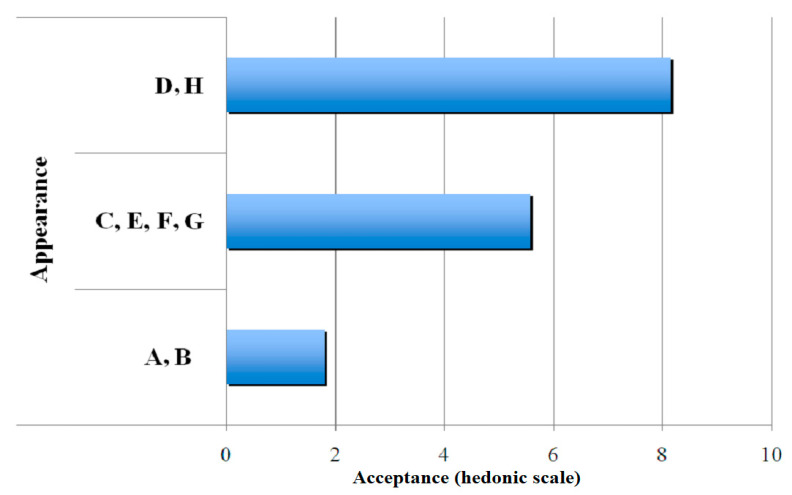
Results of clustering of gelatin-gal base samples for the attribute, appearance.

**Figure 5 polymers-13-01289-f005:**
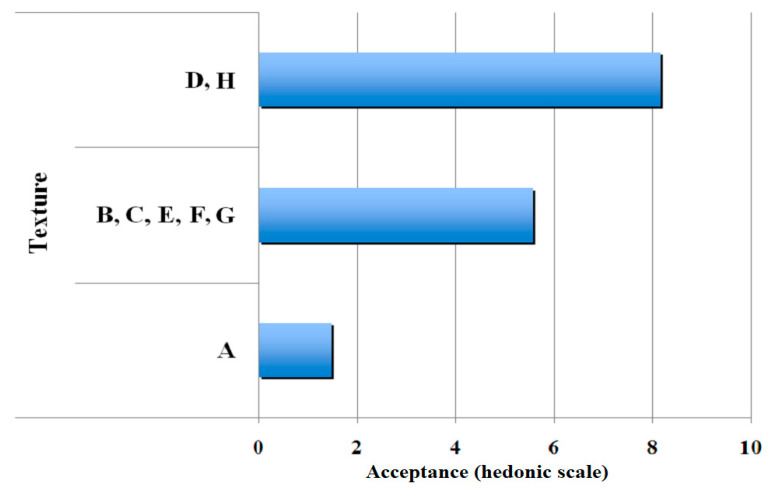
Results of clustering of gelatine samples for the attribute, texture.

**Figure 6 polymers-13-01289-f006:**
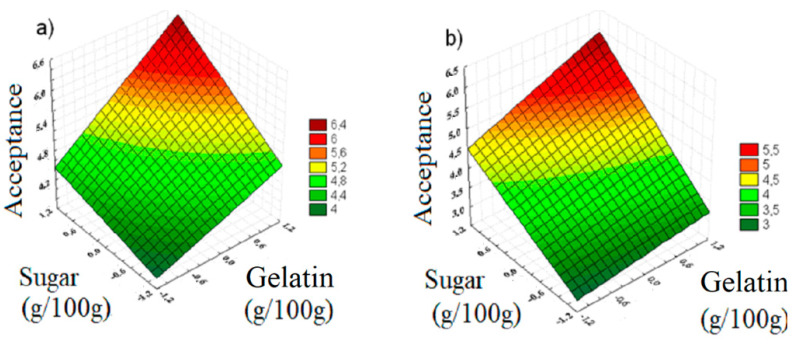
Response surfaces to optimise the gelatin-base gel attributes: (**a**) colour, (**b**) flavour, (**c**) aroma, and (**d**) texture.

**Table 1 polymers-13-01289-t001:** Level of variables of experimental design.

Gelatin-Base Gel Samples	*x* _1_	*x* _2_	Gelatin (g/100 g)	Sugar (g/100 g)
Gelatin A	−1	−1	1.903	7.1
Gelatin B	1	−1	3.806	7.1
Gelatin C	−1	1	1.903	28.6
Gelatin D	1	1	3.806	28.6
Gelatin E	0	0	2.857	17.9
Gelatin F	0	0	2.857	17.9
Gelatin G	0	0	2.857	17.9

**Table 2 polymers-13-01289-t002:** Composition analysis of powdered gelatin from chicken feet.

Composition *	(g/100 g)	Sub-Composition (g/100 g)
Moisture	9.749	
Minerals	4.807	
	Na	0.0285 ± 1.10^−4^ *
	Mg	0.0088 ± 3.10^−5^
	Ca	0.0081 ± 4.10^−5^
	S	0.0072 ± 4.10^−5^
	K	0.0041 ± 1.10^−6^
	P	0.0008 ± 8.10^−6^
	Si	0.0001 ± 8.10^−6^
Lipids	6.919	
Proteins	78.525	
	Gelatin	70.90 ± 0.52 **
Gel strength, bloom	294.79 ± 0.50 ***	
Calories (kcal)	376.37	

Obtained by ICP-MS analysis *; FTIR analysis **, and texturometrical analysis *** [4].

**Table 3 polymers-13-01289-t003:** Results of statistical analysis of gelatin qualities.

Sample	Colour ^a^	Aroma ^b^	Flavour ^c^	Texture ^d^	General Appearance ^e^
Gelatin A	3.76	2.68	2.60	2.32	2.80
Gelatin B	4.60	3.10	2.70 ^c^	5.88	3.72
Gelatin C	4.38	4.16	4.18 ^c^	3.64	3.64
Gelatin D	5.92	5.30	5.76 ^c^	6.76	6.06
Gelatin E	4.76	4.08	4.28 ^c^	5.30	4.9
Gelatin F	4.72	4.54	4.14 ^c^	4.40	4.3
Gelatin G	5.90	4.90	5.16 ^c^	6.10	5.32
Gelatin H	7.32	7.36	7.34 ^c^	7.2	7.26

Same letters means that there exist significant differences at 95% of level confidence: *t*_0.05,(98)_ = 1.66.

## Data Availability

http://www.saraujo.pro.br/sas/, accessed on 14 April 2021.

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
