# Peer review of "Combination of Computational Techniques to Obtain High-Quality Gelatin-Base Gels from Chicken Feet"

_polymers, 2021, doi:10.3390/polym13081289_

Round 1

Reviewer 1 Report

The manuscript “Combination of computational techniques to obtain high-quality gelatin from chicken feet”, by José C. C. Santana, Poliana F. Almeida, Nykael Costa, Isabella Vasconcelos, Flavio Guerhardt, Dimitria T. Boukouvalas, Wonder A. L. Alves, Pedro C. Mendoza, Felix M. C. Gamarra, Segundo A. V. Llanos, Sidnei A. Araujo, Ada P. B. Quispe, Rosangela M. Vanalle and Fernando T. Berssaneti, presents results on the optimization of the production process of gelatines from chicken feet through the use of a computational strategy. Data on the sensorial analysis was used for this purpose.

In general, the manuscript is easy to follow and read. The redaction is correct and only minor details are subject to corrections. I have included some corrections in the PDF document (attached). Perhaps may be a good idea to specify in the title or the abstract that obtained gelatins are “edible” gelatins (suggestion).

Concerning the introduction, the authors presented the relevant background for their work in the introduction. However, I think this section contains excessive information. Some journals have a specific section for “Theory” and, as far I understand, this is not the case of Polymers. Therefore, my advice to the authors would be to make a revision of the information in the introduction and make the effort to shorter this section.

The section material and methods, in general, provides enough information concerning the preparation of gelatins, the analytical methods and the computational technique. I have only a few observations about this section. I wonder if the grape flavour was the same in all preparations. The same applies to the dye added. Did the authors add a dye? Since the authors are comparing gelatin obtained from a different raw material with commercial gelatin, I believe that is important to specify that all additives were the same (except sugar, which was clearly stated its content changed). For the evaluation of attributes, did the panel have any training? Please, provide this information.

It is clear the motivation of the work. The authors suggested that other raw materials for the production of gelatin should be explored, and I agree. The authors should be more emphatic in showing the relation of their contribution to the circular economy.

In summary, this is a paper that requires some corrections and they are not critical. I believe that this paper can be considered for publication in Polymers after minor revisions.

Author Response

Response to Reviewers #1

We understand your position on the length of the introduction, since most of the authors of this article are technical scientists and we defend your position regarding short, objective introductions. But we would like to kill it because we think it is necessary to characterize the Kohonen neural network technique and its recent application in food products. This, we believe, will attract readers to our article.

On flavor - The gelatine samples with the same concentration of sugar and grape flavoring agents had the same taste, as they were made in a single batch and in the same mixer, the other gelatines were not.

Regarding the use of edible gelatin, another reviewer suggested the use of gelatin-based gels.

-------------

In article text

Abstract and methods - We had to replace the words "factorial" for "experimental" throughout the text.

Pg 2 - Brazil, as well as other countries producing poultry meat, suffer internal influences with the variation of its price in the international market, as producers concentrate on the export of these meat making them an expensive product in Brazil and the Brazilian population has low purchasing power and as such, they are unable to eat this meat, due to its high price. This is happening right now, as this (and other) meat had an increase of 3 times in its price.

Pg 5 – We had to change “petri dishes” for “Petri dishes” and “planning” for “design”.

Pg 7 - In the place of João we forget to insert the references that are at the end of the paragraph. We fixed that and we also inserted it: as well as Islam and Jewish religion

We thank you for all your attention and considerations given to our article

Reviewer 2 Report

Having read the paper I conclude that the paper is interesting for scientific and industrial readers.

Nevertheless, the manuscript needs some clarifications. Otherwise, from my point of view, it should not be published:

  1. Regarding the title of the manuscript.

“Combination of computational techniques to obtain high-quality gelatin from chicken feet“.

I have been dealing with preparation of gelatins from different poultry by-products for many years. At my first sight on the title I was 100% sure that your paper dealis with preparation of gelatins from chicken feet. This fact is also stated in Abstract “Thus, this study aimed to optimise the production of gelatin from chicken feet…” and in the last paragraph of Introduction (page 4) “This study aimed to optimise the production process of gelatins from chicken feet…” Having read the paper I found that the aim of the paper was to optimise the composition of the mixture – collagen and sugar contents – to prepare gelatin samples (A-G), see e.g. Table 1 on page 5., and test some characteristics of prepared mixtures (colour, flavor, aroma, texture). Is it true ? If yes, then I recommend to modify the title of the paper. Otherwise the readers will be confused.

  1. Materials and Methods; 2.1. Preparation of gelatins.

It is stated that chicken feet collagen powder was prepared according to the methods by Santana. OK ! Then, seven gelatin samples based on chicken feet collagen + sugar were prepared (see Table 1). OK !

I have problem to denote samples A-G as gelatins.

Gelatin is a water-soluble protein prepared from collagen tissues (pork, bovine, poultry, fish); gelatin is colourless and tasteless.

Your e A-G prepared samples should not be labeled as “gelatins”. Maybe “Gelatin-based gels” will be better.

To conclude above two points, the modified title of your paper should (e.g.) be as follows:

“Combination of computational techniques to obtain high-quality gelatin-based gels prepared from chicken feet collagen”

What do you think ?

Remember to modify in this sense the aims of the study in Abstract and Introduction as well !

Further, the last sentence of 2.1. chapter describing the extraction of collagen from chicken feet:

“Then, 200 g of chicken feet was placed in a thermal bath of 4% acetyl acid solution at 60 °C for 4 h, to extract the collagen [2, 4, 5, 11, 14]“

I have problem with the term “collagen”. I think that according to this method a gelatin (a collagen product) is extracted. This needs to be clarified. If a gel strength of 6.67 % of solution of this product was obvious and measured, then it is gelatin.

Then I would avoid using term “collagen” where it is appropriate. For example in Table 1, table 2, in the coherent text etc.

By the way, “acetic acid” is correct, not “acetyl acid”.

I reckon sequence regarding your study is as follows: raw chicken feet – separation of unwanted substances from chicken feet – chicken feet collagen – extraction of chicken feet gelatin (using 4% CH3COOH at 60 deg. C for 4 hours) – preparation of gelatin based gels (samples A-G) combining chicken feet gelatin and sugar according to RSM – testing some sensory characteristic of prepared gels (and comparing with commercial gelatin) – evaluation of the results.

  1. Regarding gel strength of gelatins.

The method described on page 6, chapter 2.4. / a) is correct.

I do not understand why gel strength is expressed in “kPa” units – chapter 3.1 in the text, Table 2. Gelatin gel strength is expressed in “Bloom” or in “g” (this is identical) units; rarely the authors use “Newtons” (but which is also OK). Please, download this and read carefully:

Standard testing methods for edible gelatin. Official Procedure of the Gelatin Manufacturers Institute of America, Inc. Available online:

http://www.gelatin-gmia.com/images/GMIA_Official_Methods_of_Gelatin_Revised_2013.pdf

  1. Conclusion

The first sentence is a bit confusing. The collocation “to compare sensory samples“ sounds illogical.

Based on my 1. and 2. remarks the first sentence of the Conclusion should be modified as well, e.g. as follows:

“The results showed that SOM is a suitable computational tool to study sensory characteristics of the samples and identify the best gelatin addition to prepare consumer acceptable gelatin-based gels.”

  1. Figure 6

The font on x and y axes is not readable. Should be improved.

As discussed earlier, instead of term “Collagen” term “Gelatin” should be used.

Generally:

Think of all my remarks deeply and revise the paper very carefully, use the correct terminology. The aim is to improve your paper as much as possible.

Author Response

Response to Reviewer #2

Regarding optimization, we used different software to optimize more than one gelatine preparation step and we were confused to present all the steps in the title, but in the summary and objective these steps are specified. This justification and its excellent modification suggestion are keys to a better understanding of our article.

Thus, regarding the title, purpose and other considerations about gelatines: we fully agree and made the changes and changes to the text and title (gelatin-base gels)

In table 1, we had to change “Samples” for “gelatin-base gel samples” to avoid having to change all the names "gelatin A, B, C ... H" over the entire length of the text and figures.

We wanted to summarize the methodology presented in Santana et al. 2020, not to be repetitive, but you perfectly described the steps of the procedure, by the way we switched to acetic acid.

Checking the journal's rules, we have opted for kPa instead of Bloom, but we changed the units in Table 2, as requested for you.

Thank you very much for your magnificent follow-up to change in conclusions.

We had improving figure 6 for a better understanding of its data

You wrote a large text, but it perfectly explained the reasons why I should change my text, because the wrong use of a word can lead to misunderstanding about my article. And, at the end of your evaluation you still end with this sentence: “The aim is to improve your paper as much as possible”. My research group and I have no words to thank you for your contribution to our article.

We thank you for all your attention and considerations given to our article

Reviewer 3 Report

Manuscript ID: polymers-1185095

Title: Combination of computational techniques to obtain high-quality gelatin from chicken feet

Authors: José C. C. Santana * , Poliana F. Almeida , Nykael Costa , Isabella Vasconcelos , Flavio Guerhardt , Dimitria T. Boukouvalas , Wonder A. L. Alves , Pedro C. Mendoza , Felix M. C. Gamarra , Segundo A. V. Llanos , Sidnei A. Araujo , Ada P. B. Quispe , Rosangela M. Vanalle , Fernando T. Berssaneti *

Presented manuscript is focused on the searching of new alternative food sources. The study develop a new source a gelatin obtained from chicken feet.

 Several remarks:

Which are the differences between samples A-G in the preparation method. Please, explain it. There is a influence of the process factors on the gelatin yield?

The FTIR analysis: if the authors can provide the graph image, it will be useful.

Composition analysis of powdered collagen from chicken feet from Table 2 must be compared and discussed with another type of collagen (commercial or manufactured).

I highly appreciate the results obtained in the present manuscript.

Author Response

Response to Reviewer #3

The process conditions (temperature, agitation and process time) for obtaining the gelatin-base gels are the same, however, as their composition is different, which makes all gelatin-base gels different (flavor, aroma, texture). One of the reviewers asked us to assume the term gelatin-based gel instead of gelatin.

Gelatins were obtained from the collagen powder of chicken feet presented in Santana et al. (2020), in which these FTIR figures are present. This article also presents the discussion and comparison of collagen with commercial gelatin.

Thus, we do not feel safe in using an new FTIR figure, although it is different figures, the images generated are similar and would lead to the questioning of this new FTIR figure being plagiarism of the figure presented in Santana 2020. We have 10 FTIR figures, but as they originated even gelatin powders and are very similar.

For this reason, we quote the reference [4] to avoid any copyright problem.

We already had this problem in a congress article and I had to send the different FTIR figures for proof and regain the confidence of the reviewer and editor. So, I apologize and please check in the article by Santana et al. 2020.

In the text above the table 2, there is a comparison of the table data with other sources of gelatins, mainly the one used (commercial) as a standard. However, we also had the following sentence inserted in the text “Chicken feet gelatin has almost two and half times more gelatin content than cowhide (~35%) [4]”.

This work is a new application of the gelatin powder extracted in “Santa et al., 2020. Valorization of Chicken Feet By-Product of the Poultry Industry: High Qualities of Gelatin and Biofilm from Extraction of Collagen. Polymers, 12, 529.

Our group also had to develop other applications of gelatine of chicken feet, as you can see in the article below:

MACIEL, C. N. ; SELLER, L. F. F. ; SOUZA, A. B. ; ALMEIDA, P. F. . Formulation of fishburgers with the addition of different protein sources and taro flour. CIÊNCIA RURAL, v. 51, p. 1-8, 2021.

ALMEIDA, POLIANA FERNANDES; LANNES, SUZANA CAETANO DA SILVA. Effects of chicken by-product gelatin on the physicochemical properties and texture of chocolate spread. JOURNAL OF TEXTURE STUDIES, v. 48, p. 392-402, 2017.

Soon, we intend to develop new products based on the research developed by our research group.

We thank you for all your attention and considerations given to our article.